# Regulation of Brain Cholesterol: What Role Do Liver X Receptors Play in Neurodegenerative Diseases?

**DOI:** 10.3390/ijms20163858

**Published:** 2019-08-08

**Authors:** Kevin Mouzat, Aleksandra Chudinova, Anne Polge, Jovana Kantar, William Camu, Cédric Raoul, Serge Lumbroso

**Affiliations:** 1Motoneuron Disease: Pathophysiology and Therapy, The Neuroscience Institute of Montpellier, University of Montpellier, Montpellier, Laboratoire de Biochimie et Biologie Moléculaire, Nimes University Hospital, 30029 Nîmes, France; 2Laboratoire de Biochimie et Biologie Moléculaire, Nimes University Hospital, University of Montpellier, 30029 Nîmes, France; 3ALS Reference Center, Montpellier University Hospital and University of Montpellier, Inserm UMR1051, 34000 Montpellier, France; 4The Neuroscience Institute of Montpellier, Inserm UMR1051, University of Montpellier, 34091 Montpellier, France

**Keywords:** Liver X receptors, cholesterol, oxysterols, neuroinflammation, amyotrophic lateral sclerosis, Alzheimer’s disease, multiple sclerosis

## Abstract

Liver X Receptors (LXR) alpha and beta are two members of nuclear receptor superfamily documented as endogenous cholesterol sensors. Following conversion of cholesterol in oxysterol, both LXR isoforms detect intracellular concentrations and act as transcription factors to promote expression of target genes. Among their numerous physiological roles, they act as central cholesterol-lowering factors. In the central nervous system (CNS), cholesterol has been shown to be an essential determinant of brain function, particularly as a major constituent of myelin and membranes. In the brain, LXRs act as cholesterol central regulators, and, beyond this metabolic function, LXRs have additional roles such as providing neuroprotective effects and lowering neuroinflammation. In many neurodegenerative disorders, including amyotrophic lateral sclerosis (ALS), Alzheimer’s disease (AD), and multiple sclerosis (MS), dysregulations of cholesterol and oxysterol have been reported. In this paper, we propose to focus on recent advances in the knowledge of the LXRs roles on brain cholesterol and oxysterol homeostasis, neuroinflammation, neuroprotection, and their putative involvement in neurodegenerative disorders. We will discuss their potential use as candidates for both molecular diagnosis and as promising pharmacological targets in the treatment of ALS, AD, or MS patients.

## 1. Introduction

Liver X Receptors (LXRs) are lipid-activated transcription factors that belong to the nuclear receptor superfamily [1]. Since their identification in the mid-1990s [2,3], they have been the focus of many studies. The initial discovery of oxidized forms of cholesterol, known as oxysterols, included natural ligands of these nuclear receptor identifying them as central actors in lipid homeostasis, particularly cholesterol homeostasis [4]. However, the growing interest for LXRs and the generation of mice with genetic ablation of *Lxrs* [5] has shed light on their pivotal role in many physiological functions, including endocrine system [6], female and male fertility [7,8], inflammation, and immune processes [9]. Pathological conditions, such as cancer, have also been studied [10]. LXR roles in the nervous system have been widely reported, and these nuclear receptors could be involved in neurodegenerative disorders [11].

## 2. LXRs: Structure and Mode of Action

LXRs exist as two distinct isoforms known as LXR alpha (LXRα) and LXR beta (LXRβ). They are encoded by two distinct genes: *NR1H3*, for LXRα, located on chromosome 11, and *NR1H2*, for LXRβ, located on chromosome 19. Although many splice variants coexist, the predominant transcripts of both genes share a similar structure. The *NR1H3* predominant transcript (NM_005693) has nine coding exons and encodes a 447-amino-acid protein, while the *NR1H2* predominant transcript (NM_007121) have eight coding exons and encodes a 463-amino-acids protein. Both proteins share an identical structure, involving four functional domains, common to typical nuclear receptors. The N-terminal domain is the most variable domain among members of the nuclear receptor superfamily. It contains regulatory regions responsible for co-activators binding, including one related to the AF-1 function. The DNA-binding domain (DBD) is a hallmark of all nuclear receptors and its sequence is highly conserved in the superfamily. LXR DNA-binding activity is mediated by two zinc-fingers, responsible for the recognition of the specific LXR Responsive Element (LXRE) within the promoter of target genes. While a common consensus response element sequence may be identified (two AGGTCA cores separated by four nucleotides [2]), some differences in DNA-binding have been observed for the two LXRs [12,13], with some promoters being preferentially bound by one of the isoforms. The DBD also contains the nuclear localization signal domain (NLS). The hinge domain has a double role, which allows flexibility of the nuclear receptor bound on DNA as well as the recruitment of co-repressors while inactivated [14,15]. The C-terminal domain has many central roles, including dimerization with the Retinoid X Receptor (RXR), recruitment of co-activators, and ligand binding. The latter function is mediated by a hydrophobic pocket that recognizes the various ligands of both receptors [6,16]. Differences in LXRα/LXRβ expression sites have been observed, but it is now admitted that all animal tissues can express both isoforms (www.nursa.org, last accessed date: 7 August 2019).

Their mechanism of action is similar to other “RXR partners” nuclear receptors. In their canonical mode of action, LXRs are constitutively bound to LXRE within the promoter of the target gene. In the absence of the activating ligand, the RXR/LXR complex recruits co-repressors, while maintaining the chromatin in a non-transcription-permissive state (i.e., “compacted state”), due to epigenetic mechanisms such as histone deacetylase activities [17,18]. The binding of an activating oxysterol and/or 9-*cis* retinoic acid (respectively LXR and RXR activating ligands) induces conformational changes responsible for the release of co-repressors and the binding of co-activators [6]. The latter can trigger epigenetic changes (e.g., Histone acetyl transferase activities), while maintaining the chromatin in a transcription-permissive state (Figure 1) [19].

The anti-inflammatory role of LXRs has been observed for a long time, as activation of LXRs by an activating ligand decreases the expression of many pro-inflammatory genes (e.g., inducible nitric oxide synthase, interleukin (IL)- IL-1β, IL-6, and tumor necrosis factor (TNF)-α) [20,21,22]. However, while the canonical mode of action of LXRs was a trans-activation mechanism, it had been postulated that this anti-inflammatory effect was indirect. A direct trans-repression mechanism involving small ubiquitin-like modifier (SUMO) proteins has been described, which explains a direct negative effect of LXRs on pro-inflammatory target genes upon LXRs activation [9,23]. Binding of a ligand to LXR could also induce its SUMOylation, which results in a cell type-specific gene inhibition mechanism. In macrophages, SUMOylated LXR stabilizes the interaction between NF-κB and co-repressors on the target-gene promoters, which, then, represses their expression. In astrocytes, target gene inhibition is mediated by preventing a signal transducer and activator of transcription 1 (STAT1) by SUMOylated LXRs [24]. However, this trans-repression mechanism was recently challenged by two recent studies. Ito et al. showed that the anti-inflammatory effect of LXRs was dependent on the canonical transactivation mechanism, involving RXR heterodimerization [25]. Additionally, this repressive effect on LPS-induced pro-inflammatory genes, including *Cox2*, *Il1*β, *Il6*, *Cxcl1*, *Ccl2*, and *Nos2*, required *Abca1* induction and the subsequent cholesterol efflux. These findings were confirmed in vivo using *Abca1^-/-^* mice. Thomas et al. confirmed these findings and showed that LXR activation reduced the chromatin accessibility in *cis* on pro-inflammatory gene enhancers containing LXRE, with an assay for transposase-accessible chromatin using sequencing (ATAC-seq) [26]. This effect was independent on AP-1 activity. Altogether, these two studies show the importance of LXR-mediated cholesterol metabolism regulation on inflammation. Further studies will be necessary to understand the relative contribution of trans-repression versus transactivation mechanisms on anti-inflammatory action upon LXR activation. Particularly, in CNS, further studies on the molecular mechanisms mediating gene inhibition in astrocytes will help to understand the link between cholesterol metabolism and neuroinflammation. In addition, non-genomic actions of LXRs have also been described, but these effects remain to be explored [9].

## 3. LXRs Ligands

As many nuclear receptors, LXRs have been first identified by studies based on sequence homology with other known nuclear receptors. Their ligands were, thus, unknown, which leads to initially consider them “orphan” nuclear receptors. The first attempt to identify LXR ligands used constructs made from the fusion of the LXRα LBD to the GAL4 transcription factor. This initial screening of more than 70 molecules led to the identification of a series of LXRα-activating oxysterols [27]. Oxysterols are oxidized-derivatives of cholesterol or cholesterol precursors [28,29]. Among the impressive number of known oxysterols, only some of them are activating LXR ligands. Particularly, 22(*R*)-hydroxycholesterol, 24(*S*),25-epoxycholesterol, or (25R)-26-hydroxycholesterol (formerly known as 27-hydroxycholesterol [30]) have been intensively studied. It had been early noticed that oxysterol profiles depend on their synthesis site. Interestingly, 24(*S*)-hydroxycholesterol has been one of the first LXR-activating oxysterol described in the central nervous system (CNS), and has been consecutively named cerebrosterol [16]. Desmosterol, which is an intermediate of cholesterol biosynthesis that can accumulate up to 30% of total brain sterols, has recently been shown as a major physiological ligand for LXRs [31,32]. The role of cholesterol in the brain has been known for decades, where its metabolism is known to be the highest compared to other tissues [33]. Thus, cholesterol is crucial for the formation and the support of myelin sheath, but also synaptogenesis, axonal plasticity, neuroprotection, and recovery following neuronal injury [34,35]. Additionally, it is now well admitted that oxysterols play a central role in the CNS [36]. 24(*S*)-hydroxycholesterol and (25R)-26-hydroxycholesterol produced either by peripheral cells or neurons can cross the blood-brain-barrier (BBB) to reach the brain, where they can interact with LXRs, or be eliminated into the blood circulation [37]. Brain 24-hydroxycholesterol has been shown to be the major source of circulating cholesterol and its plasma levels are correlated with the number of metabolically active neurons [38]. In the brain, the conversion of cholesterol into 24-hydroxycholesterol also represents the main cholesterol excretion route. The concentrations of some oxysterols, such as 24(*S*)-hydroxycholesterol and (25R)-26-hydroxycholesterol both in plasma and cerebrospinal fluid, have been linked to many neurological disorders, including neurodegenerative diseases [39]. Although these oxysterols were initially considered neuronal cholesterol by-products, more recent studies suggest a direct involvement of these molecules on CNS physiology, such as nerve impulse, neurogenesis, and memory processes [33,40,41].

## 4. LXRs: Two Central Regulators of Cholesterol Metabolism

The initial finding of oxysterols as LXR ligands logically led to study a putative role for LXRs in cholesterol metabolism. One of the first in vivo evidence of this control arouse from the generation of mice with targeted deletion of *Lxr*α [5]. Contrary to wild-type animals, *Lxr*-deficient mice were unable to stimulate *cyp7a1* gene expression in response to dietary cholesterol. This gene encodes a key enzyme in the first step of cholesterol conversion into bile acids, and, thus, their elimination from the body. This study was the first showing the in vivo role of LXRs in the control of cholesterol concentrations, particularly in the liver. Although a tight link between oxysterol and cholesterol was well admitted, many oxysterols being enzymatically synthesized from cholesterol [29], the in vivo proof that oxysterols were a direct link between cholesterol and LXRs came from the engineering of transgenic mice unable to produce 24(S)-hydroxycholesterol, 25-hydroxycholesterol, and (25R)-26-hydroxycholesterol [42]. These mice were unable to activate many LXR-target genes in response to dietary cholesterol. During the last two decades, the involvement of LXRs has been shown in almost every cholesterol metabolism pathways (for a review, see Reference [4]). Beyond the activation of the *cyp7a1* promoter, LXR-induced cholesterol efflux from the liver is also controlled by Abcg5 and Abcg8, which are two cholesterol transporters responsible for its efflux in bile [43,44]. LXRs also negatively regulate cholesterol biosynthesis by reducing the expression of enzymes involved in cholesterol *de novo* biosynthesis [45,46,47], and reverse cholesterol transport to the liver [48]. The latter includes cellular efflux from peripheral cells via Abca1, Abcg1, Abcg4, Abcg5, and Abcg8 [43,49,50,51], but also the blood transport of cholesterol toward the liver by stimulating apolipoprotein genes, such as APOE/CI/CII/CIV [52,53]. Altogether, these data enlighten the pivotal role for LXRs as an endogenous cholesterol sensor in the regulation of both intracellular cholesterol concentration, but also the control of its homeostasis within the whole body.

## 5. Cholesterol and Oxysterol in the CNS: Which Place for LXRs in the Regulation of Brain Homeostasis?

Cholesterol is one of the major constituents of the CNS. The brain contains approximatively 25% of whole body’s cholesterol [54]. In the CNS, the synthesis rate of cholesterol is also the highest with 95% of brain cholesterol content being produced de novo [33,55]. It is a key constituent of myelin, where it accounts for 70% of brain content, and neural cell membranes [34]. At a cellular level, cholesterol has been shown to promote synaptogenesis, axonal plasticity, neuroprotection, and glial cells’ proliferation [56]. In adult stages, the main source of neuronal cholesterol arises from astrocytes, neuronal cholesterol biosynthesis being relatively low [33]. The brain-derived neurotrophic factor (BDNF) exerts several protective roles within the CNS, including neuronal survival, differentiation, and plasticity. In cultured neurons, BDNF treatment reduced cholesterol uptake [57]. This effect was associated with an increase in LXRβ expression, which suggests a protective role for this nuclear receptor against an excess of neuronal cholesterol. Hence, based on this well-established role for cholesterol in the CNS, it appears that its regulation would be essential to maintain CNS normal functions [58]. LXRs have been shown to modulate brain cholesterol homeostasis at various stages. The neuronal cholesterol concentrations have, thus, been demonstrated to be modulated at three levels: (1) cholesterol uptake by neurons is negatively regulated by LXRs, via the degradation of the LDL receptor (LDLR) by an inducible degrader of LDLR (IDOL) [11], (2) LXR activation by a synthetic agonist stimulated neuronal cholesterol efflux [59], and (3) LXRs control the cholesterol supply from astrocytes to neurons. The latter pathway is now known as the main source of cholesterol for neurons [33]. Thus, both (24S)-hydroxycholesterol and GW683965A (a synthetic LXR-agonist) up-regulate ABCA1 and ABCG1 in astrocytes, which promotes cholesterol efflux from this cell type. APOE expression is also increased, which mediates the cholesterol transport toward neurons [60]. In oligodendrocytes and Schwann cells, which provide cholesterol to the myelin sheath, LXR can control both cholesterol homeostasis and myelination processes [61,62].

LXRs could also exert a neuroprotective role in the CNS, which acts on neuroinflammation. In a mouse model of intracerebral hemorrhage, brain lesions were associated with an increase in LXRα production [63]. LXR induction, with the widely used synthetic agonist T0901317, also attenuated tissue loss and neuronal death, BBB, and brain edema, microglial activation, and pro-inflammatory responses. Microglia considered as the resident immune cells of the CNS, are involved in many essential functions including phagocytosis, elimination of toxics, synaptic pruning, and optimization of neurotransmission [64]. These cells account for 0.5% to 16.6% of CNS cells, depending on the anatomical region [65]. Microglial activation leads to the acquisition of the M1 phenotype, which is a pro-inflammatory phenotype that is a hallmark of many neurodegenerative diseases, such as Alzheimer’s disease (AD), Parkinson’s disease, Amyotrophic Lateral Sclerosis (ALS), and Frontotemporal dementia (FTD). In recent years, the use of in vitro models revealed that activation of LXRs can reduce many pro-inflammatory factors produced by cultured glial cells, including iNOS and nitric oxide, COX-2, IL-1β, IL-6, and MCP-1 and impaired the IκB-NF-κB pathway [66,67,68]. Microglial activation was also reduced in various in vivo mouse models upon treatment with LXR agonists [63,69]. Taken together, these findings strongly support a key role for LXRs in the CNS homeostasis, both in the regulation of brain cholesterol metabolism and in the control of microglia-dependent neuroinflammatory processes. Further studies of LXRs in cellular and molecular mechanisms underlying neurodegenerative diseases may help in the future to understand the natural history of these diseases.

## 6. What Role for LXRs in Neurodegenerative Diseases?

Since cholesterol and oxysterol homeostasis are crucial to a correct development and maintenance of CNS functions, and LXRs are central regulators of their metabolism, it is tempting to suggest that dysregulations of LXRs could be involved in the risk of developing neurological diseases, and/or in the modulation of their natural history. In this study, we review the putative involvement of these nuclear receptors in several neurodegenerative disorders. In three of them, the targeted activation of LXRs could be a goal to achieve in the future: ALS, AD, and Multiple Sclerosis (MS).

### 6.1. Amyotrophic Lateral Sclerosis

Amyotrophic Lateral Sclerosis (ALS) is a rapidly fatal adult-onset neurodegenerative disease characterized by the progressive death of both upper and lower motor neurons (MN). Mean age at onset is 65 years and median survival time after onset is 3 years [70]. The incidence of ALS is 2-3/100 000 and overall prevalence worldwide is 7-10 /100 000 [71,72]. The clinical consequences of the MN degeneration in ALS is a progressive paralysis, starting by one leg or speech, and spreading on the body leading to tetraplegia, anarthria, aphagia, and ending with respiratory failure. Cognitive impairment is not rare in ALS patients, and 20%–50% of the patients exhibit front-temporal lobe dementia (FTD) [73]. Almost 10% of ALS cases inherited (FALS) with a prominent autosomal dominant trait. In FALS cases, pathogenic variants in many genes have been described, but variants within four major genes have been identified: *C9ORF72*, superoxide dismutase-1 (*SOD1*), TAR DNA binding protein (*TARBP*), encoding the protein TDP43, and fused in sarcoma (*FUS*) [74]. Repeat expansions in *C9ORF72* is the most frequent genetic abnormality accounting for more than 40% of FALS cases and, overall, for 3% to 4% of ALS cases [75,76].

Pathophysiology of ALS is not fully understood but clues may come from the high variability of the prognosis of ALS patients ranging from a few months to more than 40 years, which may suggest that prognosis, and pathogenesis, are linked to the diffusion of a biological agent, in a prion-like mechanism [77]. Many different cellular pathways have been implicated in the pathogenesis of ALS such as impairment of RNA metabolism, protein aggregation, cellular and nuclear transport dysfunction, and glutamate excitotoxicity [78]. In addition, neuron-microglia interactions seem central in ALS pathogenesis, and CNS immune cells infiltration [79], as well as microglia and astrocyte activation, together with MN degeneration, are among hallmarks of the disease [58,80].

Cholesterol and oxysterol metabolism have been described as impaired in ALS patients, but this is still debated. Some studies showed elevated plasma cholesterol in ALS patients compared to controls, while others did not [81,82]. In a cohort of French patients, an elevated LDL/HDL ratio was associated with better prognosis [83]. Elevated cholesterol has been observed in cerebrospinal fluid (CSF) of ALS patients [84]. Analysis of cholesterol metabolites also showed defects in brain cholesterol and bile acid metabolism. Recently, serum and CSF levels of 25-hydroxycholesterol were shown to be higher in ALS patients than in controls, and the levels were also associated with disease severity [85].

Based on the well-known roles of LXRs in the control of cholesterol metabolism and in the regulation of immune processes, mainly by their anti-inflammatory action, LXRs seem to represent good candidates to be studied in ALS pathophysiology. The use of *Lxr* deficient mice by Gustafsson’s group revealed a motor phenotype comparable to ALS patients’ symptoms. *Lxrβ^-/-^* mice had progressive impaired motor performance leading to hind limb paralysis, associated with motoneuron degeneration and loss of neuromuscular junctions [86,87,88]. The animals also had signs of astrocytic activation and elevated pro-inflammatory factors in the spinal cord. Ubiquitin and TDP-43 inclusions in motoneurons were noted, which is one hallmark of brain pathology in ALS patients [89]. Oral administration of β-sitosterol, which is a putative analog of a neurotoxic plant sterol [90], worsened the ALS-like phenotype of lxr*β*^-/-^ mice. This suggests a neuroprotective role for the nuclear receptor. This sterol also impaired 24-hydroxycholesterol metabolism, bringing evidence that LXRβ could act as a regulator of brain cholesterol and oxysterol metabolisms, with this role being, at least in part, important for its neuroprotective role.

A proteomic analysis of sera from ALS patients and controls revealed significant changes in many protein concentrations [91]. Among them, a bioinformatic analysis showed the LXR/RXR pathway as one of the most significant. Recently, we showed for the first time that both LXRα and LXRβ were genetic modulators of the ALS phenotype in patients [92]. In a cohort of 438 patients and 330 healthy controls, two LXRα SNPs were shown to be associated with age at onset. Particularly, age at onset was increased for heterozygous patients (+3.9 years) and was much higher in patients carrying the homozygous rare genotype (+7.8 years) for the LXRα SNP rs2279238 (*p* = 0.003, false discovery rate (FDR) = 0.012). The prognosis of ALS patients was influenced by the LXRβ genotype. The LXRβ SNP rs2695121 was associated with the duration time of the disease. T/T carriers had a significantly shorter survival time than C carriers (Hazard Ratio = 1.47 [1.12–1.93], *p* = 0.0055, FDR = 0.044). Although no unbiased strategies such as gene wide association studies pointed an association between these genes and ALS, this study encourages the further study of the participation of LXRs to MN death in ALS.

Altogether, many studies today point toward a neuroprotective role for LXRs, which makes them good candidates for both molecular diagnosis and strengthens the need for searching *NR1H3* and *NR1H2* variants responsible for the disease and/or its prognosis, as well as pointing to their future use as pharmacological targets in the treatment of ALS patients.

### 6.2. Alzheimer’s Disease

AD is the most common form of dementia, characterized by progressive memory loss, aphasia, apraxia, and agnosia, due to degeneration of cortical neurons. Most AD cases begin after 70 years old. Almost 44 million people suffer from dementia worldwide, and AD accounts for 50% to 75% of them [93,94]. Around five to seven million new cases of AD are recorded each year. Pathologically, the disease is characterized by the presence of brain amyloid deposits, neurofibrillary tangles, and neuronal loss [95]. One of the major constituents of these deposits is the Aβ peptide, which arises from the sequential cleavage of the amyloid precursor protein (APP). The genetic basis of the disease is complex. Three major genes, called *APP*, *PSEN1*, and *PSEN2* (encoding Presenilin-1 and 2, respectively), may be responsible for early onset AD cases, but only account for less than 1% of all cases. Several genetic susceptibility factors have been described, but the genetic landscape of the disease remains largely elusive. It is noteworthy that one of the major genetic risk factors for AD is the well-known LXR-target APOE, where the APOEε4 allele increases the risk of developing the disease from three times (for heterozygous carriers) to 12 times (for homozygous carriers).

The link between cholesterol and AD has been suggested when the treatment of patients with statins (HMG-CoA reductase inhibitors) were associated with a lower prevalence of AD [96,97]. In mice models of AD overexpressing mutated APP, the administration of T0901317 lowered both soluble Aβ40 and Aβ42 [98]. In these animals, genetic loss of either LXRα or LXRβ worsened brain deposits [99]. The activation of LXRs has several biological consequences: (1) In cultured glial cells, it attenuates the inflammatory response induced by fibrillary Aβ, (2) In AD mouse models, it induces not only overexpression of the LXR-targets ABCA1, ABCG1, and APOE, but also reduces cognitive defects and improves brain pathology [100,101]. The anti-inflammatory role of LXRs in the CNS, particularly on microglia, has also been shown with, when activated, a reduced production of pro-inflammatory factors particularly involved in the NF-κB pathway [102]. In primary cultures of hippocampal neurons, administration of the synthetic LXR-agonist GW3965 prevented synaptic defects induced by Aβ exposure [103]. In AD mice models, GW3965 modulated synaptic plasticity in vivo and reduced astrogliosis [100].

In humans, the progression of brain atrophy has been associated with a decrease in circulating concentration of 24-hydroxycholesterol [38], which suggests a link between the disease and oxysterols’ metabolism. The LXR ligand desmosterol has also been studied in AD, as DHCR24, which is the enzyme catalyzing the conversion of desmosterol to cholesterol, was suggested to be down-regulated in brains of AD patients. Although still a matter of debate, these original findings led to propose Selective Alzheimer’s Disease Indicator-1 (Seladin-1) as an alternate name for this enzyme [104]. Beyond this controversy, it is interesting to notice that brain desmosterol levels are decreased in AD patients compared to age-matched controls [105]. The first genetic link between LXR-encoding genes and AD has been found in 2006, with a mild association of a haplotype comprising rs1405655 and rs2695121 within the LXRβ-encoding gene (*NR1H2*) [106]. The association of a genetic interaction between *NR1H2* and *CD14* genes with AD further suggested a possible link between LXRβ and the modulation of immune processes in the disease [107]. The LXRα-encoding gene has also been implicated since rs7120118 within *NR1H3* was associated with AD [108]. The C allele of this single nucleotide polymorphism (SNP) was also associated with a decrease of Tau and phospho-Tau in the CSF of the patients, whose increase is known to be a hallmark of AD. The same group reported a significant association of rs7120118 CC but not the CT genotype with a decrease in soluble Aβ42 concentration in the temporal cortex of patients compared to the TT genotype [109]. The CT genotype, but not the CC genotype of the same SNP, was also associated with LXRα expression, whereas LXRs targets ABCA1, ABCG1, and APOE expressions were not influenced by the rs7120118 genotype.

Several results are in favor of a protective role of both LXR isoforms in humans and for the involvement of LXRs in the molecular mechanisms underlying AD pathogenesis. While their exact role remains to be uncovered, the data show that the pharmacological activation of LXRs could be considered a potential therapeutic target in AD.

### 6.3. Multiple Sclerosis

MS is a neuroinflammatory disorder characterized by late, progressive neurodegeneration [110]. MS is one of the most prevalent chronic neurologic disorders in young adults, with a highly variable prevalence, ranging from less than 1 to more than 100 per 100,000 people per year [111]. There is also a large variability in age of onset, varying from early childhood to more than 70 years, but mean age of onset is 31 years [112]. To date, no clear monogenic cause has been identified, but many susceptibility loci have been identified [113]. The most admitted genetic association is an increased risk of MS in the HLADRB1*1015 allele carriers. Correlatively, pathogenesis of MS clearly involves autoimmune and neuroinflammatory processes in the CNS. Typically, the symptoms are represented by relapses during which patients develop neurological deficits in relation to brain inflammatory foci secondary to blood brain barrier (BBB) lesions. This condition is called relapsing remitting MS (RRMS). If relapses are not prevented efficiently by the treatment, a second, progressive phase may occur after a variable delay, marked by a relentlessly progressive handicap. This is the Secondary Progressive MS (SPMS). However, almost 10 of the patients never experience relapses but have, since the onset a progressive handicap, experienced primary progressive MS (PPMS). In RRMS, neuroinflammatory mechanisms are largely prominent and the efficiency of the various treatments for this condition have demonstrated that. In SPMS and PPMS, neurodegenerative processes are involved and anti-inflammatory drugs are not efficient.

Immune processes are a well-known LXR-controlled pathway. Cholesterol turnover has long been suggested as involved in neurological diseases [114,115]. Side-chain oxysterols, particularly 24-hydroxycholesterol and (25R)-26-hydroxycholesterol, have been proposed to reflect cerebral cholesterol turnover. Their use as diagnostic markers has, hence, been studied as these side-chain oxysterols, which can cross the BBB. In patients with MS, there is a decrease in circulating 24(S)-hydroxycholesterol and this decrease is associated with the severity of the disease [116]. Various oxysterol profiles have been observed depending on the type of MS [117,118]. Myelin is one important target of the inflammatory process in MS. LXRs have been shown to regulate myelination and remyelination, by modulating expression of several genes involved in myelin sheath formation [61,119].

In patients, although no SNP within LXR-encoding genes has been associated with MS, a missense mutation in the *NR1H3* gene (NM_005693.3:c.1244G>A; p.Arg415Gln; rs61731956) in families with PPMS was recently described [120]. The variant segregated with the disease in two unrelated families. Haplotype analysis of the variant carriers suggested that they might arise from a common ancestor. Although predicted pathogenic by in silico analyses, incomplete penetrance was reported, which supports the need for additional genetic analyses. This finding, however, has been a source of debate [121]. A study from The International Multiple Sclerosis Genetics Consortium did not show any association of this variant with MS risk in a wider population comprising 32,852 controls and 36,000 MS cases (assumed to be 13-fold larger than Wang et al. population) [122]. No association with the disease in the PPMS subpopulation could be found, nor any association with an MS clinical course. Another group argued against pathogenicity of the variant using variant frequencies available in public databases [123]. Nevertheless, this issue remains in dispute [124]. This enlightens one of the most important challenges in human genetics known as variant interpretation. The wide use of next-generation sequencers in human molecular diagnosis today leads to the identification of many variants, whose role in human pathology may be difficult to interpret. Although unique efforts in the standardization of variant interpretations have been made [125], there is a need for new tools to help the classification of variants of uncertain significance, especially in neurodegenerative diseases.

## 7. Conclusions

In recent decades, cholesterol and oxysterol metabolism has demonstrated their pivotal role in maintaining brain homeostasis. In several neurodegenerative disorders, cholesterol and oxysterol dysregulation were frequently observed, which raises the question of their putative use as biomarkers. LXRs, well known as cholesterol endogenous sensors, have proven their key role as CNS cholesterol gatekeepers. These fascinating nuclear receptors must not be reduced to their cholesterol-lowering function. Once activated, they act as anti-inflammatory actors, especially in the brain. We can, thus, propose a dual role for LXRs, both maintaining brain cholesterol metabolism, as well as repressing neuroinflammatory processes. Studies have shown their involvement in many neurodegenerative disorders, such as ALS, AD, and MS (Figure 2).

In vitro, in vivo, and human studies all showed the involvement of LXRs in neuroprotection. Right now, those works led us to a key point. Since they are inducible transcription factors, the development of pharmacological agonists targeting LXRs for treating neurodegenerative disorders have to be considered.

## Figures and Tables

**Figure 1 ijms-20-03858-f001:**
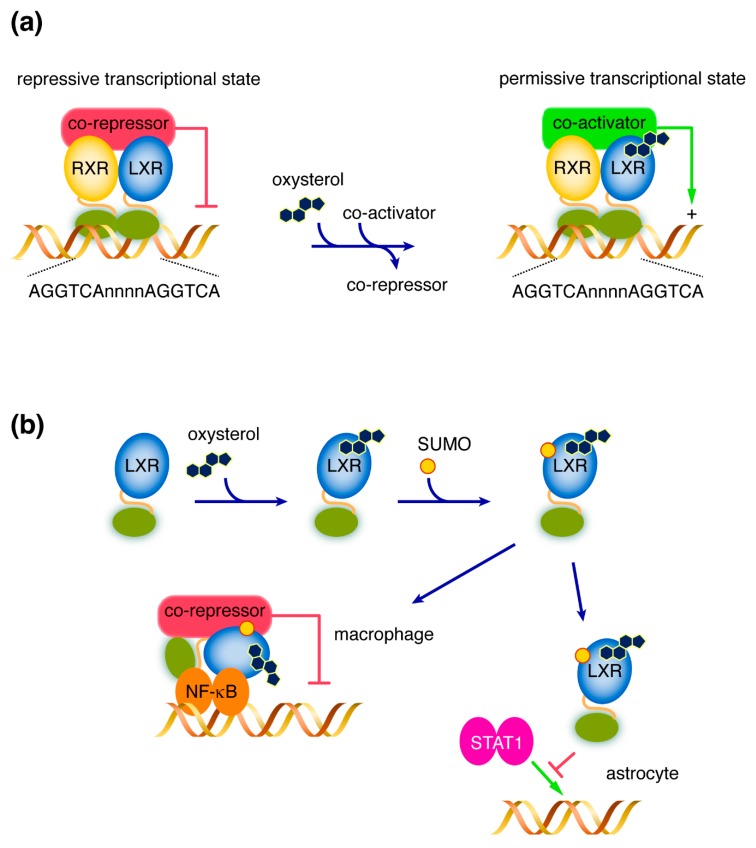
Schematic representation of the LXR mode of action. (**a**) In their canonic mode of action, the RXR/LXR heterodimer is constitutively bound to DNA on its response elements (LXRE—whose consensus sequence is AGGTCAnnnnAGGTCA). In the absence of the ligand, the heterodimer interacts with co-repressors, maintaining the chromatin in a repressive transcriptional state. The binding of either an RXR or an LXR ligand (e.g., an oxysterol) induces conformational changes, which leads to the release of co-repressors and the recruitment of co-activators, and promotes a chromatin permissive transcriptional state and the activation of target genes. (**b**) In the trans-repression mechanism, binding of an LXR-ligand induces its SUMOylation. In macrophage, SUMOylated LXR stabilizes co-repressors on NF-κB, which, therefore, down-regulates the expression of the target gene. In astrocytes, SUMOylated LXR prevents target gene expression by blocking the binding of STAT1 to promoters. This trans-repression mechanism is a source of debate and its in vivo contribution to anti-inflammatory response still has to be confirmed. Green arrow: activation; red T bar: inhibition. LXR, Liver X Receptor. LXRE, LXR Responsive Element. RXR, Retinoid X Receptor. STAT1, signal transducer and activator of transcription 1. SUMO, small ubiquitin-like modifier.

**Figure 2 ijms-20-03858-f002:**
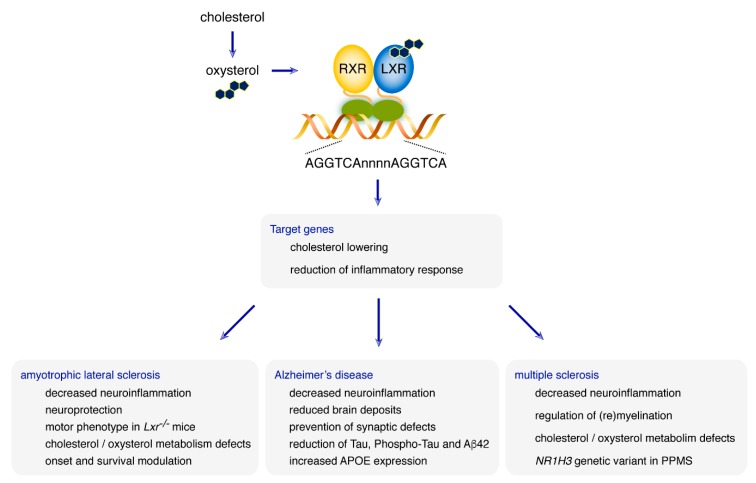
Involvement of LXRs in neurodegenerative diseases. LXRs are central regulators of both cholesterol and oxysterol metabolisms. They have multiple roles in brain functions. Dysregulations of LXR-controlled metabolisms have been observed in neurodegenerative disorders. In amyotrophic lateral sclerosis, animal studies suggested that LXRs decrease neuroinflammation and act as neuroprotectors. *Lxr*-defective mice also have a motor phenotype mimicking ALS. In patients, cholesterol and oxysterol metabolism disturbances have been observed and LXR encoding-genes polymorphisms are associated with age at onset and survival time. In Alzheimer’s disease, treatment of AD animal models with an LXR agonist decreases neuroinflammation and reduces brain deposits. The cell culture and animal models also show that LXR could prevent synaptic defects. In patients, LXRα encoding-gene polymorphism is associated with a decrease in Tau and phospho-Tau levels in the CSF of patients, as well as in Aβ42 in the temporal cortex. *APOE*, whose allele ε4 is strongly associated with AD, is also an LXR-target gene. In multiple sclerosis, LXRs have been shown to promote myelination and remyelination and to decrease neuroinflammation. In patients, cholesterol and oxysterol metabolism have been observed and a missense genetic variant has been described as responsible for familial forms of primary progressive MS. LXR, Liver X Receptor. PPMS, Primary Progressive Multiple Sclerosis. RXR, Retinoid X Receptor.

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
