# Peer review of "Regulation of Brain Cholesterol: What Role Do Liver X Receptors Play in Neurodegenerative Diseases?"

_ijms, 2019, doi:10.3390/ijms20163858_

Round 1

Reviewer 1 Report

This is a well written and interesting review regarding the potential role of LXRs in neurodegenerative diseases.

I have the following suggestions :

The anti-inflammatory roles of LXRs are central in many of the processes discussed by the authors regarding neurodegenerative diseases. The section discussing the mechanisms involved (lines 89-101) could be completed:

The sumoylation mechanism has been described in 2007 (not so recently), however very recent publications have challenged the relative contribution of this mechanism to the overall anti-inflammatory activity of LXRs and suggested that cholesterol efflux may play a prominent role. (Ito A, Hong C, Rong X, Zhu X, Tarling EJ, Hedde PN, Gratton E, Parks J, Tontonoz P., LXRs link metabolism to inflammation through Abca1-dependent regulation of membrane composition and TLR signaling. Elife. 2015 Jul 14;4:e08009. doi: 10.7554 ;  Thomas DG, Doran AC, Fotakis P, Westerterp M, Antonson P, Jiang H, Jiang XC, Gustafsson JÃ…, Tabas I, Tall AR. LXR Suppresses Inflammatory Gene Expression and Neutrophil Migration through cis-Repression and Cholesterol Efflux. Cell Rep. 2018 Dec 26;25(13):3774-3785). These references should be briefly discussed.

The original study regarding the sumoylation of LXR and STAT1 in astrocytes should be referred (Lee JH, Park SM, Kim OS, Lee CS, Woo JH, Park SJ, Joe EH, Jou I., Differential SUMOylation of LXRalpha and LXRbeta mediates transrepression of STAT1 inflammatory signaling in IFN-gamma-stimulated brain astrocytes. Mol Cell. 2009 Sep 24;35(6):806-17.)

Recently, desmosterol has been described as a major physiological ligand of LXRs by Christopher Glass group. This should be included in the section regarding the LXR ligands. Is there any evidence of an alteration of desmosterol levels in the brain during neurodegenerative diseases?  This could be an interesting point to discuss.

The genetic associations between LXRs and the risk of some neurodegenerative diseases discussed by the authors have been obtained by targeted candidate-gene approaches. Is there any evidence obtained by unbiased strategies such as gene wide association studies ? This point could be discussed.

Author Response

We first would like to thank the reviewer for these constructive comments. Here is our point-by point response:

The anti-inflammatory roles of LXRs are central in many of the processes discussed by the authors regarding neurodegenerative diseases. The section discussing the mechanisms involved (lines 89-101) could be completed:

The sumoylation mechanism has been described in 2007 (not so recently), however very recent publications have challenged the relative contribution of this mechanism to the overall anti-inflammatory activity of LXRs and suggested that cholesterol efflux may play a prominent role. (Ito A, Hong C, Rong X, Zhu X, Tarling EJ, Hedde PN, Gratton E, Parks J, Tontonoz P., LXRs link metabolism to inflammation through Abca1-dependent regulation of membrane composition and TLR signaling. Elife. 2015 Jul 14;4:e08009. doi: 10.7554 ;  Thomas DG, Doran AC, Fotakis P, Westerterp M, Antonson P, Jiang H, Jiang XC, Gustafsson JÅ, Tabas I, Tall AR. LXR Suppresses Inflammatory Gene Expression and Neutrophil Migration through cis-Repression and Cholesterol Efflux. Cell Rep. 2018 Dec 26;25(13):3774-3785). These references should be briefly discussed.

Answer: According to reviewer’s comment, we added a new paragraph discussing these two interesting references (lines 102-115), as well as a new sentence within Figure 1 legend (lines 84-86). “Recently” (line 92) has also been removed.

The original study regarding the sumoylation of LXR and STAT1 in astrocytes should be referred (Lee JH, Park SM, Kim OS, Lee CS, Woo JH, Park SJ, Joe EH, Jou I., Differential SUMOylation of LXRalpha and LXRbeta mediates transrepression of STAT1 inflammatory signaling in IFN-gamma-stimulated brain astrocytes. Mol Cell. 2009 Sep 24;35(6):806-17.)

Answer: Thank you for your comment. The reference has been added.

Recently, desmosterol has been described as a major physiological ligand of LXRs by Christopher Glass group. This should be included in the section regarding the LXR ligands. Is there any evidence of an alteration of desmosterol levels in the brain during neurodegenerative diseases?  This could be an interesting point to discuss.

Answer: Desmosterol has been described in the ligand section as a major LXR physiological ligand (lines 129-131). Interestingly, this sterol accounts for up to 30% of total brain sterols. The contribution of desmosterol to AD has also been discussed (lines 311-316). Unfortunately there is no evidence of alteration of desmosterol levels in brain of ALS or MS patients.

The genetic associations between LXRs and the risk of some neurodegenerative diseases discussed by the authors have been obtained by targeted candidate-gene approaches. Is there any evidence obtained by unbiased strategies such as gene wide association studies ? This point could be discussed.

Answer: To our knowledge, no GWAS studies revealed any association between LXR-encoding genes and ALS, AD or MS. This point has been discussed in ALS section, as it is the only disease with a SNP association discussed (lines 274-276).

Reviewer 2 Report

ijms-568374

Comments:

The review by Mouzat et al. describes in a very concise way the links between Liver X Receptors, as master regulators of cholesterol metabolism, and three main neurodegenerative diseases.

I guess it’s time for the scientific community to become aware that the nomenclature of 27-hydroxycholesterol is wrong and that, in accordance with the IUPAC rules, the chemical name of this oxysterol must be (25R),26-hydroxycholesterol. Thus, the Authors should have to modify accordingly, by citing Fakheri’s paper (“27-Hydroxycholesterol, does it exist? On the nomenclature and stereochemistry” Steroids 77 2012, 575).

Some flaws referring to English were found throughout the paper, which negatively affects the quality of the work. Just some examples:

Page 1 line 36, these nuclear receptors Page 1 line 38, “shed light” , not “shed the light” Page 1, lines 38-40, the Authors should rewrite this sentence: It is not possible cite “cancer” among physiological functions; and is the nervous system a physiological function? 4 line 109 “of one the first “should be “one of the first”. 4 line 114 “The role of cholesterol has been known for decades in brain” should be “ The role of cholesterol in the brain has been known for decades”. Page 5 line 155 What does it mean “metabolic rate of cholesterol”? Page 6 line 213 What does it mean “many genes have been described”? Could be “alterations in the expression of many genes” more suitable? Page 6 line 241 “a putative neurotoxic plant sterol” The reference has to be included. Page 7 line 246, Delete “In ALS patients” because just later “...sera fom ALS patients”. Page 7, line 269, remove capital letter from amyloid. Page 7, line 283, delete “an” between “and” and “improves”. Page 8, line 313, Remove “to date” or “so far”.

Author Response

We first would like to thank the reviewer for these constructive comments. Here is our point-by point response:

I guess it’s time for the scientific community to become aware that the nomenclature of 27-hydroxycholesterol is wrong and that, in accordance with the IUPAC rules, the chemical name of this oxysterol must be (25R),26-hydroxycholesterol. Thus, the Authors should have to modify accordingly, by citing Fakheri’s paper (“27-Hydroxycholesterol, does it exist? On the nomenclature and stereochemistry” Steroids 77 2012, 575).

Answer: Thank you for this comment. The first occurrence of 27-hydroxycholesterol has been replaced by “(25R)-26-hydroxycholesterol (formerly known as 27-hydroxycholesterol)” and Fakheri’s paper has been cited. All other occurrences of 27-hydroxycholesterol have been replaced by (25R)-26-hydroxycholesterol (lines 123-124).

Some flaws referring to English were found throughout the paper, which negatively affects the quality of the work. Just some examples: Page 1 line 36, these nuclear receptors Page 1 line 38, “shed light” , not “shed the light”

Answer: This error has been corrected.

Page 1, lines 38-40, the Authors should rewrite this sentence: It is not possible cite “cancer” among physiological functions; and is the nervous system a physiological function?

Answer: Thank you for your helpful comment. The sentence has been rewritten as following: “However, the growing interest for LXRs and the generation of mice with genetic ablation of Lxrs [5] has shed light on their pivotal role in many physiological functions, including endocrine system [6], female and male fertility [7,8], inflammation and immune processes [9]. Pathological conditions, such as cancer, have also been studied [10]. LXR roles in the nervous system have been widely reported, and these nuclear receptors could be involved in neurodegenerative disorders [11].“

4 line 109 “of one the first “should be “one of the first”. 4 line 114 “The role of cholesterol has been known for decades in brain” should be “ The role of cholesterol in the brain has been known for decades”. Page 5 line 155

Answer: These errors have been corrected.

What does it mean “metabolic rate of cholesterol”?

Answer: Thank you for this comment. As it was not clear, “Metabolic rate” has been replaced by “the synthesis rate”.

Page 6 line 213 What does it mean “many genes have been described”? Could be “alterations in the expression of many genes” more suitable?

Answer: The sentence has been replaced by: “In FALS cases, pathogenic variants in many genes have been described, but variants within 4 major genes have been identified”

Page 6 line 241 “a putative neurotoxic plant sterol” The reference has to be included.

Answer: Kim et al (reference 89) actually postulated that b-sitosterol was a putative neurotoxic plant sterol. As this was not clear, we modified the sentence as “Oral administration of b-sitosterol, a putative analog of a neurotoxic plant sterol” and added the original reference showing that β-sitosterol-β-d-glucoside is neurotoxic (reference 90).

Page 7 line 246, Delete “In ALS patients” because just later “...sera fom ALS patients”. Page 7, line 269, remove capital letter from amyloid. Page 7, line 283, delete “an” between “and” and “improves”.

Answer: These errors have been corrected.

Page 8, line 313, Remove “to date” or “so far”.

Answer: “So far” has been removed